# Comprehensive Analysis of Industrial Solid-Waste-to-Energy by Refuse-Derived Fuel Technology: A Case Study in Shanghai

**Ting Li [1], Wei Li [1], Ziyang Lou [2],\* and Luochun Wang [1]**

1   School of Environmental Chemistry and Engineering, Shanghai University of Electric Power, Shanghai 201306, China; 18222183075@126.com (T.L.); dr_weili@foxmail.com (W.L.); wangluochun@shiep.edu.cn (L.W.)
2   School of Environmental Science and Engineering, Shanghai Jiao Tong University, Shanghai 200241, China
\*   Correspondence: louworld12@sjtu.edu.cn; Tel.: +86-13564381973

**Abstract:** The prolific generation of industrial solid waste (ISW) in China, coupled with its complex composition, presents significant challenges due to exceeding environmental capacity. Identifying an appropriate approach to maximize the use of ISW, particularly low-value industrial solid waste (LISW), is crucial for addressing environmental issues. This study explores the potential of converting LISW into refuse-derived fuel (RDF), an energy-rich precursor, as a promising method for disposal and reutilization. The advantages of RDF lie primarily in two key areas: management and technology. Regulatory aspects cover principles governing RDF feedstock preparation, storage and transportation requirements, and pollutant emission regulations. Technical considerations include pretreatment techniques, additive selection, and analyzing RDF as a substitute for fossil fuels. To assess the effectiveness of RDF technology in harnessing the remaining energy from LISW, this paper provides an overview of relevant national laws and regulations concerning incineration plants, guiding the utilization of RDF in such facilities. Additionally, using Shanghai as a case study, we evaluate the ISW situation, domestic waste incineration plants, and cement kiln plants to identify potential scenarios for RDF application in future energy systems. Our findings suggest that LISW holds significant potential as a power plant fuel, particularly when blended with higher calorific value materials to produce RDF particles with exceptional combustion performance, density, and storage characteristics.

**Keywords:** low-value industrial solid waste; refuse-derived fuel; waste-to-energy

## 1. Introduction

The generation rate of Industrial solid waste (ISW) has been steadily increasing, surpassing the capacity of the natural environment and municipal authorities to respond due to the technological advancements and changes in consumption patterns [1]. According to the "China Statistical Yearbook-2019", China's ISW generation reached a staggering 3.3 billion tons in 2017, with Shanghai leading the nation in production at 16.3 million tons [2]. Environmental concerns surrounding ISW have spurred the development of new strategies for energy production. Governments worldwide, including the U.S. Environmental Protection Agency, recognized ISW as a source of renewable energy as early as 2015 [3]. The Chinese government has increased environmental investments in industrial sectors related to solid waste, with the allocation rising from 9.72% to 23.53% [4].

Incineration has emerged as a highly promising choice for the industrial production sector in terms of energy, economics, and environmental considerations, making it a core WTE technology [2]. Modern incineration plants employ combustion conditions that ensure the complete mineralization of organic matter, including persistent organic pollutants (POPs). In the 1980s, the emergence of RDF injected vitality into the energy conversion of LISW, coinciding with the global energy crisis of the 1970s [5].

RDF could contribute to significant waste volume reduction and energy recovery, and some industries are increasingly interested in RDF as a fuel substituted source, including

cement and power industries, along with industrial boilers. Energy-intensive industries like cement, paper, chemicals, and power generation in Germany are exploring RDF for co-combustion in existing plants or dedicated boilers [6]. RDF is a standardized fuel with low moisture content, minimal pollutants, and a high calorific value derived from various waste sources [7]. Cement kilns offer a suitable application for RDF and reduced energy consumption and carbon dioxide emissions [8,9], and RDF incineration offers superior efficiency in waste reduction and environmental impact [10], which has been widely employed in the United States and Europe [11]. The current state of RDF technology and the legislative challenges associated with alternative fuels [12], and the economic analyses [13] were reviewed. The replacing coal with RDF have achieved significant reductions in $CO_2$ emissions (up to 40%), with RDF derived from rejected municipal solid waste fractions [14].

There is a general lack of comprehensive analysis, particularly regarding RDF production from LISW in China. Chinese government has made significant strides in promoting the development of the waste incineration industry, involved implementing a series of policies and enacting legislation to incentivize waste incineration projects in major cities. The objective of this study is to provide a comprehensive review of RDF preparation technologies, management policies, and relevant laws and regulations in American and European countries. The aim is to explore the potential of utilizing LISW as a secondary fuel to produce RDF and to assess the feasibility of implementing RDF technology in Shanghai, China. This research fills a critical gap by examining the legal, technological, and practical aspects of RDF in China. This comprehensive approach aligns with the growing societal demand for effective resource utilization and sustainable development practices.

## 2. Status of RDF

### 2.1. Management Requirements

#### 2.1.1. Properties of RDF

RDF holds significant importance as an alternative energy source. When assessing the viability of RDF as a fuel, the calorific value emerges as the pivotal parameter [14]. The utilization of RDF necessitates adapting to the combustion process, thereby emphasizing the significance of its calorific value in determining the overall quality of RDF. Several additional parameters warrant consideration as well. Numerous countries select RDF raw materials based on their value level and availability. Fixed carbon, moisture content, volatile content, and elemental composition adhere to specific standard requirements [15]. Various standards exist for the preparation of RDF. The current RDF standard, developed by the European Association of Specialized Waste Heat Treatment Companies in Belgium, serves as an alternative fuel for waste incineration in cement plants [16]. A summary of the standard is presented in Table 1, illustrating the inclusion of several fundamental parameters for quality evaluation [14]. These disparities arise from variations in parameter inclusion and the prescribed limit values for each parameter. Italy's standards primarily focus on proximate analysis and heavy metal elements, while the parameter setting in Spain is relatively less comprehensive compared to EURITS and Switzerland.

**Table 1.** RDF preparation standards for some European countries.

| Parameter | Unit | General Standards | | Cement Kiln Standards | | |
|---|---|---|---|---|---|---|
| | | IT | FI | EURITS | CH | ES |
| Moisture | % | <25 | | | <10 | <1 |
| Calorific Value | KJ/kg | 15 | | 15 | 25.1–31.4 | |
| Ash content | % | 20 | | 5 | 0.6–0.8 | <10 |
| Cl | % | 0.9 | 0.01 | 0.5 | <1 | |
| S | % | 0.6 | 0.01 | 0.4 | <0.5 | 3 |
| N | % | | 0.01 | 0.7 | | 3 |
| Na + K | % | | 0.01 | | | |
| Al | % | | 0.01 | | | |
| Hg | mg/kg | | 0.1 | | 5 | |
| Cd | mg/kg | | 0.1 | | 5 | |
| Pb | mg/kg | 200 | | | 100 | 2500 |
| Cu | mg/kg | 300 | | | | |
| Mn | mg/kg | 400 | | | | |
| Cr | mg/kg | 100 | | | 30 | 1500 |
| Zn | mg/kg | 500 | | 500 | 2000 | |
| Ni | mg/kg | 40 | | | 10 | |
| As | mg/kg | 9 | | | | |
| Cd + Hg | mg/kg | 7 | | | | |
| Br/I | mg/kg | | | 0.01 | | |
| Hg/Ti | mg/kg | | | 2 | | |
| As, Se, Cd, Sb | mg/kg | | | 10 | | |
| Mo | mg/kg | | | 20 | | |
| V | mg/kg | | | | 50 | |
| Zr | mg/kg | | | | 300 | |
| Halogens | mg/kg | | | | | 5 |

The selection of the appropriate RDF necessitates a comprehensive evaluation of various forms, ranging from liquid to solid and from powder to lumpy. Ensuring a flexible fuel supply is essential, whether it involves direct feeding into the combustion area or integration into the preheating system [17,18]. The chosen RDF should follow the principles of cost-effectiveness, ease of handling, convenience in storage, and an extended storage life, which will be shown belows.

2.1.2. Principles of RDF Storage

Due to the heterogeneous composition and preparation methods of RDF, it can no longer be considered a "stable" material due to its reaction and degradation rates. Storage-related issues may arise from incomplete packaging or suboptimal outcomes. Unpleasant odorous liquids may escape into the surrounding environment, and the leakage of pollutants can lead to an increase in harmful incidents. As new RDF plants continue to be established, the production capacity of RDF in many countries is rapidly expanding. The European Union alone produces an estimated 4–5 million tons of RDF annually [19,20]. The substantial quantity of RDF generated presents significant challenges in terms of storage and transportation logistics. To provide clarity on the licensing requirements for RDF storage, the Scottish Environmental Protection Agency (SEPA) released a note in September 2014 [21]. Similarly, the South Australian Environmental Protection Agency issued relevant guidelines in February 2010 to clarify the responsibilities of all stakeholders involved in RDF transportation, with the aim of mitigating the risk of accidents. Figure 1 illustrates the storage principles in Scotland and the transportation principles in South Australia. SEPA's RDF storage principles include the following: RDF stored at the dockside must be securely stored for a maximum of five days, with a total storage capacity not exceeding 4000 tons. RDF must be stored in bales wrapped to a sufficient standard that prevents water ingress, access by pests, release of odors, or escape of waste material. The South Australian guidelines outline transshipment principles that stipulate RDF transportation

must be carried out by qualified and approved professionals. It is also essential to transport RDF in suitable vehicles to reception and combustion facilities approved for this purpose.

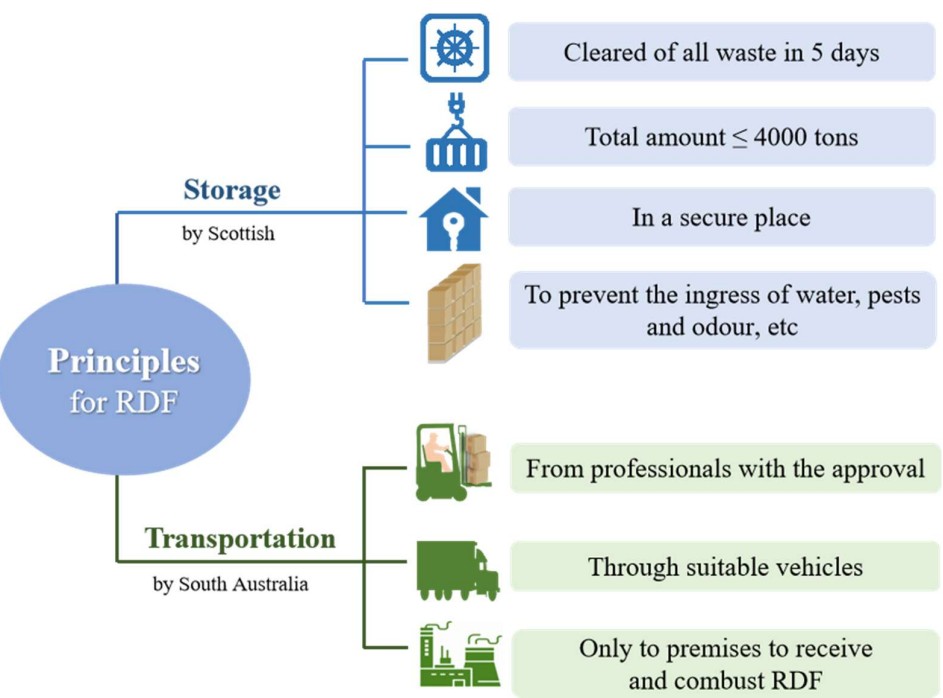

**Figure 1.** The storage principle in Scotland and the transportation principle in South Australia.

### 2.1.3. Pollution Control of RDF Burning

RDF offers several advantages over raw solid waste, including a higher calorific value and greater physical and chemical uniformity, leading to superior performance in pollution mitigation. However, it is crucial to acknowledge and address persistent environmental concerns. During ISW treatment, various harmful exhaust gases can be produced from organic waste under specific moisture and temperature conditions. Similarly, combusting particulate solid waste can lead to the dispersion of pollutants into the atmosphere, causing significant air pollution in surrounding areas. Studies indicate that air pollution from RDF combustion can become a regional issue, especially near densely populated and industrialized urban areas with RDF facilities. The significant presence of chlorine, nitrogen, and sulfur in ISW contributes to the generation of harmful gases like hydrogen chloride (HCl), sulfur oxides (SOx), and nitrogen oxides (NOx) during combustion [22]. Air quality at the RDF plant site is significantly impacted by the dispersion of air pollutants resulting from RDF combustion and prevailing meteorological conditions. Consequently, controlling pollution emissions from RDF has become a central focus of current research. For instance, introducing biomass like wood chips and rice into sewage sludge RDF has been shown to improve combustion characteristics and reduce sulfur dioxide ($SO_2$) emissions [17]. Since RDF technology falls under the waste incineration category, regulations for RDF pollution emissions heavily rely on existing standards and guidelines for waste incineration. Chronological summary of relevant legal policies [23–32] are established by various countries (Figure 2). The aforementioned laws provide a comprehensive overview of the pollutants emitted during RDF combustion (Table 2). A comparison of regulations across different regions reveals that the European Union imposes the most stringent standards on pollutant types, with a particular emphasis on controlling heavy metal emissions from flue gases. China, on the other hand, focuses primarily on limiting the total emission of heavy metal pollutants like chromium (Cr), cobalt (Co), and nickel (Ni). Notably, the EU goes a step further by specifying detailed quotas for each specific heavy metal category. In contrast, the United States exhibits less stringent regulations in this particular area.

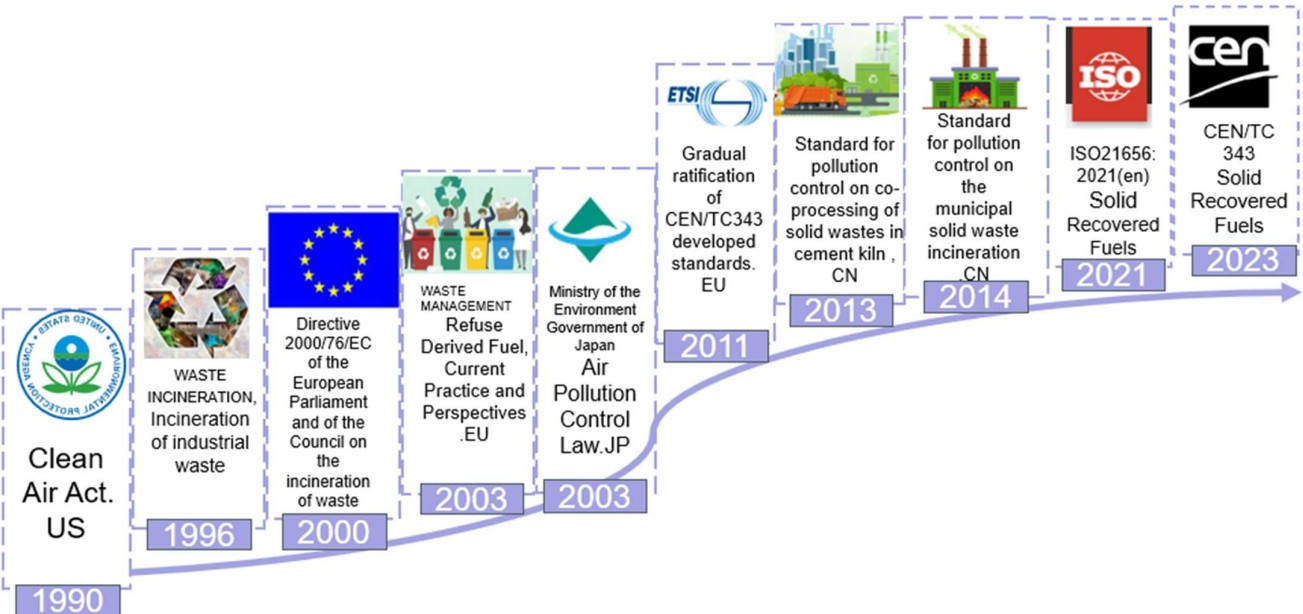

**Figure 2.** Relevant legal policies formulated by various countries in chronological order.

**Table 2.** National limits on emissions from combustion in RDF incinerators.

| Parameter (CN) | Value (CN) | Value (EU) | Value (US) |
|---|---|---|---|
| Particulates | 20 mg/m$^3$ | | 24 mg/dscm |
| NOx | 250 mg/m$^3$ | 2500 g/t | 150 ppmv |
| SO$_2$ | 200 mg/m$^3$ | 70 g/t | 30 ppmv |
| HCl | 50 mg/m$^3$ | 105 g/t | 25 ppmv |
| Hg | 0.05 mg/m$^3$ | 3 g/t | 80 µg/dscm |
| Cd + Tl | 0.1 mg/m$^3$ | | |
| Cr + Co + Ni + Cu + Mn + Pb + Sb + As | 1 mg/m$^3$ | | |
| CO | 80 mg/m$^3$ | 125 g/t | 100 ppmv |
| Dioxins | 0.1 ngTEQ/m$^3$ | | 0.2 ngTEQ/m$^3$ |
| Mn | | 0.4 g/t | 0 |
| Zn | | 21 g/t | |
| Co | | 0.3 g/t | |
| As | | 0.05 g/t | |
| Cr | | 0.3 g/t | |
| Ni | | 0.1 g/t | |
| Pb | | 35 g/t | 200 µg/dscm |
| Cu | | 3 g/t | |
| Cd | | 3 g/t | 20 µg/dscm |
| NMVOC | | 7400 g/t | |
| PAH | | 0.02 g/t | |
| Opacity | | | 0.017 min.avg. |

### 2.2. Technology of RDF Manufacturing

RDF plays a critical role as an alternative fuel for power plants traditionally reliant on coal [33]. The production process involves several stages, including size reduction, separation, crushing, drying, and pelletization. RDF is primarily produced from waste materials like cardboard, paper, various plastics, glass, and metallic and non-metallic materials. Initially, Fluff-RDF was popular due to its ease of homogenizing municipal waste. However, in Japan, the term RDF now primarily refers to Pellet-RDF. The solid fuel conversion system varies depending on the waste material but generally involves four fundamental processes: crushing, screening, drying, and shaping. The drying process can be conducted before or after shaping, and in cases with low moisture content, the drying and crushing steps may be omitted. To enhance retention and reduce hydrogen chloride gas emissions, additives are now incorporated. Additionally, specific amounts of plastic can be included as binders to improve formability, leading to the development of

various mixing methods. According to the ASTM classification standard, municipal solid waste (MSW) is classified and crushed to produce RDF2/3, which is further compressed to obtain RDF5. Chemical treatment prior to shaping is a common practice.

### 2.2.1. Pretreatment Technology

The primary objective of RDF pretreatment is to ensure its uniform size and maintain a high and consistent calorific value (ranging from 3000 to 6000 kcal/kg, depending on the waste source, typically two-thirds of standard coal). The advantages of preprocessing RDF raw materials are evident. Preprocessed RDF exhibits low pollution levels and does not release unpleasant odors. Moreover, due to its processing, RDF undergoes a significant size reduction, shrinking to one-tenth of its original size, which enables convenient storage at room temperature for 6 to 12 months without any degradation. These attributes make RDF a viable option as a primary fuel or as a blend with other fuels [34]. The fundamental aim of an efficient pretreatment process is to optimize the utilization of RDF in subsequent processes [35]. RDF pretreatment involves both physical and chemical methods, each possessing its own advantages and drawbacks, as elaborated below. Various physical pretreatment techniques, including milling, chipping, grinding, freezing, and radiation, are employed for treating different types of waste [36]. Montingelli et al. conducted a study to evaluate the effects of battering, ball milling, and microwave pretreatment on the conversion of macroalgae into biomass fuel [37]. The results indicate that the battering pretreatment yielded superior outcomes compared to the other methods. Hwang et al. utilized a 5.6 mm mesh to crush and remove industrial solid waste (ISW) from the automotive industry [38]. The floating and sinking separation technique effectively removes a significant amount of heavy fractions and incombustible materials, proving to be a simple and effective method for ash removal, thereby enhancing the fuel quality for utilization. Joseph et al. applied physical pretreatment techniques, including separation, grinding, and sieving, to samples obtained from a British waste management company to obtain fuels with different particle size ranges [20]. The findings demonstrate that physical pretreatment significantly reduces the mass and volume of raw waste. Fuels with a particle size range of 0.5–2 mm are particularly suitable for fast pyrolysis equipment used for energy recovery. These methods effectively disrupt the physical and chemical bonds of the raw materials and substantially reduce the crystallinity of macromolecules. However, one drawback of physical pretreatment is that certain methods may not be effective when used alone and often require combination with other pretreatment techniques.

Chemical pretreatment is a widely adopted approach for RDF pretreatment. Various chemical pretreatment methods, including acid pretreatment, alkaline pretreatment, oxidative pretreatment, and ionic liquid pretreatment, have been employed [37]. Acid pretreatment and alkaline pretreatment are the most commonly used methods, whereas oxidative pretreatment and ionic liquid pretreatment are limited due to their high operational costs associated with expensive chemical solvents and ionic liquids. Based on published articles and data collected from RDF facilities, acid pretreatment has demonstrated better performance when applied to biomass-based RDF [39]. On the other hand, alkaline pretreatment is found to be more suitable for ISW-type RDF pretreatment [40]. For instance, sewage sludge, a prevalent form of ISW, undergoes various processes to be transformed into RDF [41]. Since sewage sludge contains a significant amount of organic matter, predominantly proteins and carbohydrates, the pretreatment process plays a crucial role. These organic materials account for approximately 90% of the volatile suspended solids in the sludge. Yuan et al. concentrated sludge at 4 °C for 24 h at different pH values ranging from 4.0 to 11.0 to produce fuel [42]. The results revealed a considerable increase in yield under alkaline conditions. Chemical pretreatment can effectively handle complex compositions in LISW (in low-income countries' solid waste). However, chemical pretreatment also exhibits certain drawbacks, including cumbersome operation procedures and potential introduction of unpredictable pollution.

### 2.2.2. Choice of Additives

The incorporation of additives in RDF has been found to significantly enhance its calorific value while effectively mitigating the pollution resulting from RDF incineration [43]. Additives serve two main functions, depending on their objectives: capturing heavy metal compounds and purifying the combustion-generated pollution gases. Several studies have investigated the effectiveness of adsorbents mixed directly with fuel for capturing heavy metal compounds. For instance, ammonium sulfate has been demonstrated to be effective in capturing Cr, Cu, and Hg, while halloysite is effective in capturing Cd, Co, V, and Mn. Similarly, kaolinite has shown efficacy in capturing Pb [44]. Li et al. have shown that fly ash and bottom ash can adsorb and react with $SO_2$ over a wide temperature range (30–600 °C) [45]. Corella et al. studied the performance of CaO and MgO in removing HCl during the gasification of chlorinated (RDF) raw materials [46]. The emissions of RDF combustion fumes and heavy metals are directly linked to air and soil pollution, underscoring the significance of pollution reduction. Consequently, investigating the reduction in pollutants through the addition of additives during the preparation process has become a prominent research direction. Published papers on additives can be broadly classified into two categories: purification of pollution gases and capture of heavy metals. Table 3 provides a summary of recent additive types and their corresponding targets.

**Table 3.** The types of additives used in recent years and the target of the additives.

| Function | Addition Agent | Target | Year | Reference |
|---|---|---|---|---|
| Purify pollution gas | $Ca(OH)_2$ | HCl | 2008 | Chiang et al. [47] |
| | Limestone | HCl | 2005 | Partanen et al. [48] |
| | Limestone | HCl, $SO_2$ | 2005 | Partanen et al. [49] |
| | Mn-based adsorbent | $SO_2$ | 2020 | Xuan et al. [50] |
| | Carbonaceous adsorbent | $NO_2$ | 2011 | Hofman et al. [51] |
| | Waste char | NO | 2019 | Ulusoy et al. [52] |
| | Sulfates/Sewage sludge | PCDD/Fs | 2013 | Åmand et al. [53] |
| | Lime mud | $CO_2$ | 2013 | Sun et al. [54] |
| Capture heavy metal | Alumina | Pb, Sn, Cu, Zn | 2017 | Saini et al. [55] |
| | Kaolin | Pb | 2014 | Wang et al. [56] |
| | Kaolinite, Bauxite | Pb, Cd | 2024 | Liu et al. [57] |
| | $Al_2O_3$ | Cu, Zn, Pb | 2017 | Saini et al. [55] |
| | $Al_2O_3 + SiO_2 + CaO$ | PbO | 2016 | Wang et al. [58] |
| | Ammonium sulfate | Cr, Cu | | |
| | Halloysite | Cd, Co, V, Mn | 2019 | Jagodzińska et al. [44] |
| | Kaolinite | Pb | | |

Bosmans et al. [59] provide a comprehensive review of thermochemical conversion techniques employed for energy recovery from waste, whether it is in its "fresh" state or has been previously landfilled. While certain waste treatment processes accept raw municipal solid waste (MSW) as input, most advanced waste treatment technologies require pre-treated forms of MSW. This is because the chemical composition and fluid dynamics of these advanced technologies are highly sensitive to variations in feedstock, such as changes in composition, humidity, ash content, particle size, density, and reactivity. Utilizing a more uniform waste input helps to minimize variable and unstable operating conditions as well as fluctuations in product quality. Moreover, the uniform characteristics of the pre-treated waste enable more stringent process control, thereby facilitating the attainment of stricter product quality specifications.

### 2.2.3. Application of the RDF Principle

RDF is a type of fuel that is produced through the shredding, drying, and palletization of combustible waste materials such as paper, wood, plastic, leather, and textiles. This fuel can be utilized in various ways to generate energy, including conventional coal-powered power plants. RDF could be used in specialized boilers for municipal public facilities, such as those used for cold spring heating and school heating, which could be used in urban areas for regional heating and cooling purposes.

It could be also utilized in the cement industry and some special filters for agricultural equipment, such as automation warm water circulating systems. Various types of solid

fuel-fired boilers are also available: for RDF, such as Bulkhead stoker combustion system: rotary flow combustors: fluidized bed combustion systems:which have been proven and implemented in Europe and the USA [60].

*2.3. Discussions of RDF Technologies and Systems*

One of the key advantages of utilizing RDF pellets is their high calorific value (0.145 kW/kg) [61]. Coal, being a non-renewable resource with associated ash and greenhouse gas emissions when burned in thermal power plants, is contrasted by RDF, which represents a relatively clean technology that employs municipal household waste as a renewable feedstock. This approach reduces reliance on landfills, addresses waste management issues, and ultimately mitigates the environmental impact of waste. It is estimated that 750 tons of waste can yield 192 tons of RDF. When combusted in boilers, RDF can generate temperatures of up to 1600 °C, which can be utilized to produce steam and potentially generate electricity. Additionally, the resulting fly ash can find applications in the cement industry for cement manufacturing [62]. The power generation efficiency achieved through RDF is estimated to be approximately 18%, with an energy recovery rate of 168 kWh [63]. However, it is important to acknowledge that the RDF process also presents certain drawbacks, including the costs associated with land acquisition, pre-treatment equipment, and site modifications. Labor costs constitute a significant portion of the annual operating expenses, contributing to a higher net unit cost per ton compared to other waste incineration technologies.

There is a growing global trend in renewable energy production, with a particular emphasis on waste-to-energy (WTE) technology. In the Kingdom of Saudi Arabia (KSA), RDF has emerged as a promising option with higher growth potential and priority over incineration due to its superior process efficiency and reduced environmental pollution. However, the initial capital and operational costs, as well as the requirement for skilled labor, pose limitations to the widespread adoption of RDF. Prior to the establishment of incineration plants in Saudi Arabia, it is crucial to address the treatment of air and waterborne pollutants, as well as the management of ash within the incineration facilities. It is projected that by 2035, the annual power generation potential for these two scenarios will increase to 1447 MW and 699.76 MW, respectively. These figures highlight the immense potential for electricity generation through these waste disposal technologies. If properly developed, these technologies can not only promote the use of renewable energy for electricity generation but also address the cost and environmental concerns associated with landfills [33].

## 3. The Future of RDF in Shanghai

*3.1. ISW Status in Shanghai*

The rapid pace of economic growth and industrialization has resulted in a significant increase in the generation of industrial solid waste (ISW) [64]. Compared to developed countries such as the United States, Germany, Sweden, and Japan, China still has a substantial journey ahead in terms of effectively managing ISW through recycling, treatment technologies, and management strategies [65].

The European Union's experience underscores the importance of prevention and treatment in mitigating industrial solid waste (ISW) pollution. The waste management hierarchy, as depicted in Figure 3, emphasizes the 4R approach (reduce, reuse, recycle, recover) to establish the priorities for waste management. Various energy recovery processes can be employed for ISW, such as biological methods like anaerobic digestion, as well as thermal treatments including combustion, gasification, and pyrolysis [66]. Furthermore, the utilization of treated LISW in the production of RDF can be a viable destination for such waste. The introduction of RDF technology in this context enables energy recovery from the processed LISW, specifically through heat recovery. This approach serves as an effective waste recovery solution, aligning with sustainable waste management practices.

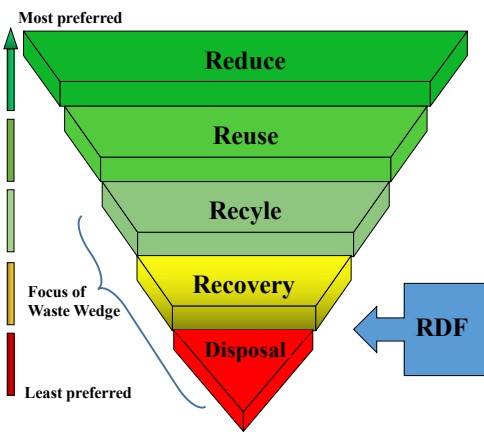

**Figure 3.** Waste management guidelines.

A comprehensive overview of the disposal, comprehensive utilization and generation volume of ISW in China over the past several years are illustrated in Figure 4 [67]. Landfilling remains the predominant method of waste disposal in China, with an apparent gap in the incineration rate when compared to developed countries. Figure 5 provides a summary of waste disposal methods employed by major countries worldwide. Notably, Norway, Denmark, and Japan exhibit higher incineration rates of 50%, 54%, and 79%, respectively. Japan, in particular, stands out as a leading proponent of waste incineration, leveraging advanced technologies and environmentally friendly incineration equipment. Given the mounting pressure on LISW treatment, incineration-based waste disposal has emerged as a priority. It is imperative to consider the environmental impact and employ environmentally friendly advanced incineration plants while ensuring economic viability. As a significant waste producer, China's waste management efforts should focus on improving the disposal of LISW to alleviate the burden of unprocessed LISW dumping. Simultaneously, considering China's energy scarcity, waste-to-energy (WTE) technologies based on RDF offer a promising solution to meet energy demands. These technologies have garnered significant attention due to their higher energy recovery rates and minimal land occupation.

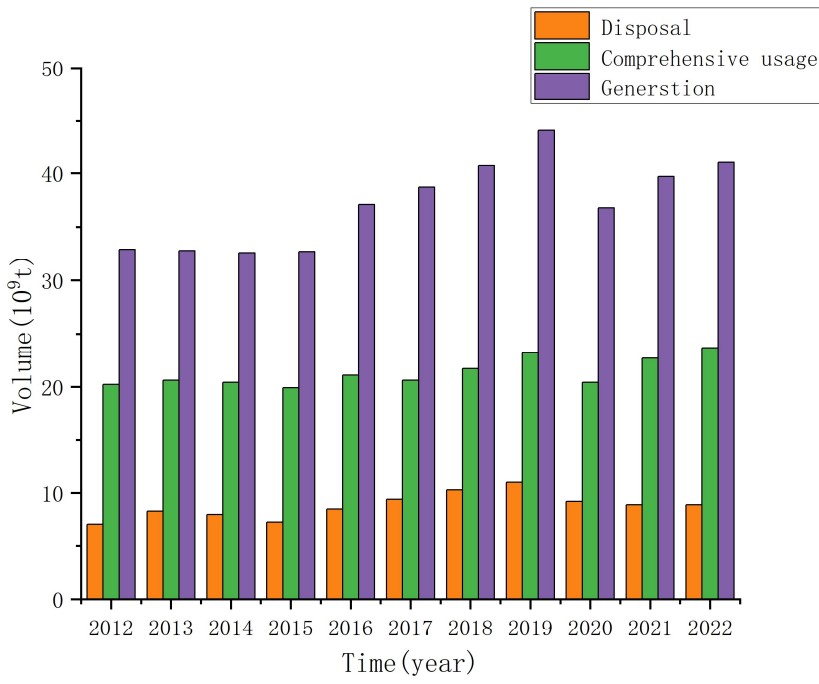

**Figure 4.** ISW treatment in China from 2012 to 2022.

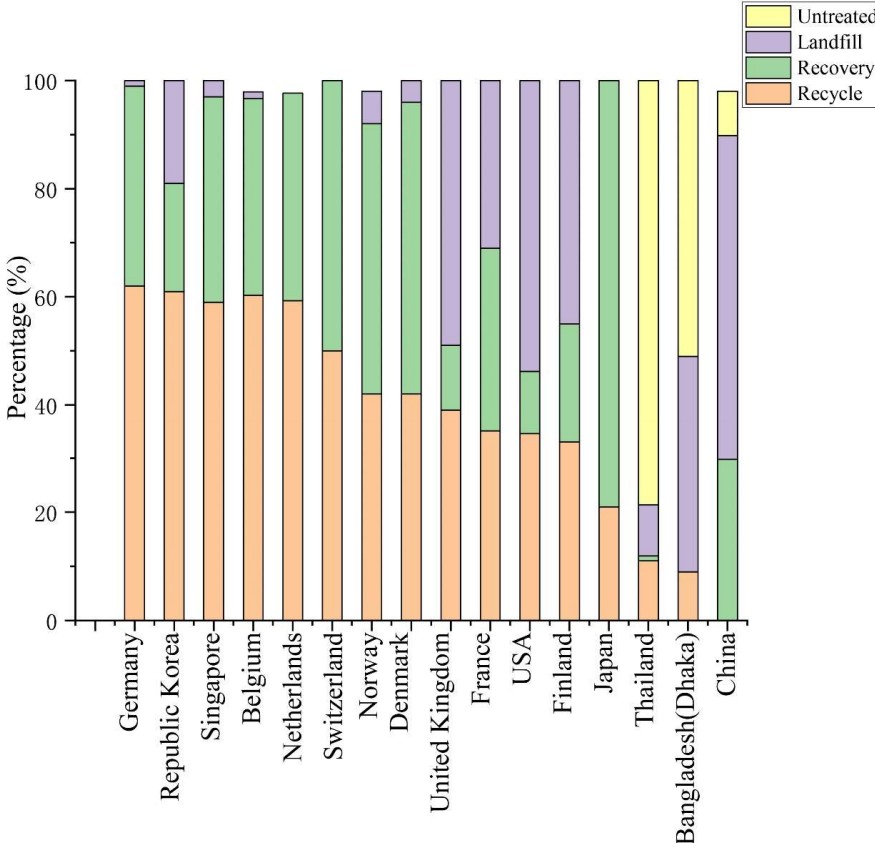

**Figure 5.** In 2017, waste management methods in different regions.

### 3.2. Heat Treatment Conditions in Shanghai

Shanghai, as China's largest and most developed city, faces a particularly severe low-calorific industrial solid waste (LISW) challenge compared to other regions [68]. However, Shanghai has taken proactive steps in addressing this issue by advocating for incineration as a primary disposal method, which has proven to be highly effective in recent years. In 2018, Shanghai's total population reached approximately 24.24 million, and its gross domestic product reached RMB 3.8 trillion in 2019 [2]. With its significant population and advanced economy, Shanghai has emerged as one of the major producers of industrial solid waste (ISW). In 2017 alone, Shanghai generated 16.3 million tons of ISW, with 15.33 million tons being comprehensively utilized. The city has dedicated disposal and storage capacities of 1 million tons and 20,000 tons, respectively [68]. Shanghai has already established itself as a leading ISW treatment area in China. However, there is still an annual surplus of over one million tons of ISW that remains underutilized, necessitating immediate measures to address this challenge. According to the "3R" principle, landfilling should be minimized and considered as the least preferable option for handling industrial solid waste (ISW) in Shanghai. Unfortunately, landfilling remains the predominant method used in the city, which is not conducive to sustainable development and the optimal utilization of land resources. Shanghai can learn from the experiences of the European Union (EU) in this regard. The EU has made significant efforts to convert waste from landfills into valuable resources. As a result, per capita landfilling in Europe has declined by 60% over the past 20 years, dropping from 830 g per capita per day in 1995 to 320 g per capita per day in 2016 [69]. To reduce reliance on landfills, there has been a growing global emphasis on energy recovery through waste-to-energy (WTE) technologies, which help prevent the waste of energy-rich materials. In 2019, the "Regulations on the management of domestic garbage in Shanghai" were officially implemented [70]. Shanghai took the lead as a pilot city in China in enforcing mandatory waste classification. The primary objective of waste classification is to categorize waste as valuable resources, and a more

detailed waste classification system facilitates the utilization of rear-end resources [71]. The separation of wet garbage from dry garbage significantly enhances the calorific value of the dry garbage fraction. The separation of wet garbage resulted in a reduction of dry garbage to 15,275 tons per day in 2019, marking a 26% decrease from 2018 [72]. From a waste classification standpoint, the calorific value of the incoming materials for incineration power plants has increased, with an adequate design margin. The mixed incineration of low-calorific industrial solid waste (LISW) following waste classification has emerged as a prominent topic in China, exemplifying Shanghai's commitment to establishing a "zero-waste city" and promoting a positive movement. Furthermore, Shanghai boasts a well-established network of heat treatment facilities. Figure 6 displays the locations and names of the nine incineration plants situated in Shanghai. Presently, these incineration plants collectively process approximately 20,000 tons of municipal solid waste daily, generating power and accounting for 71.4% of the city's daily municipal solid waste, thereby providing a foundation for the mixed combustion of LISW [73]. When incorporating LISW into the mix, it becomes challenging to meet the strict heating value requirements set by each power plant [74]. To address this challenge, the utilization of RDF technology has played a crucial role in enabling LISW to meet the specific demands of power plants. RDF allows for the blending of LISW with household waste, resulting in a denser fuel with increased mass and energy density, achieved by utilizing a relatively small amount of energy [75]. This increased density and reduced volume of LISW not only reduces transportation costs but also enhances its characteristics, making it more efficient in energy generation when carefully considered.

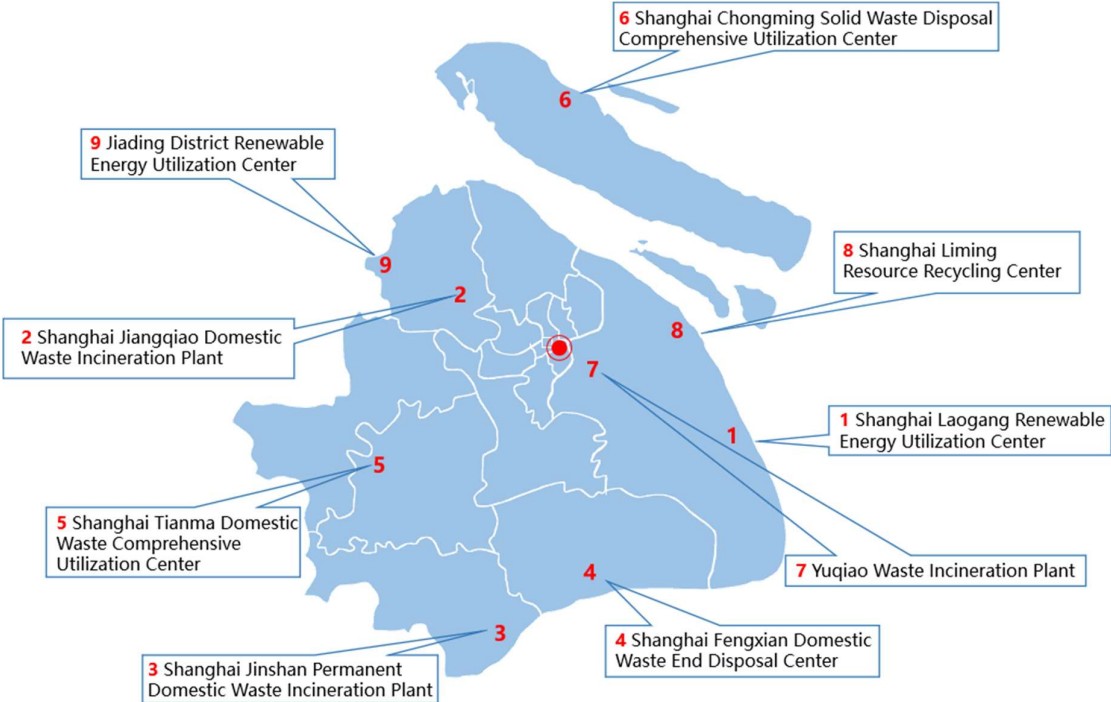

**Figure 6.** The locations of 9 incineration plants in Shanghai.

Shanghai's thriving construction industry contributes to the growing demand for cement, which heavily relies on the consumption of fossil fuels. Consequently, the cement industry not only incurs high costs but also poses significant environmental pollution challenges [76]. In fact, statistics reveal that the cement industry alone accounts for approximately 5% of the world's total $CO_2$ emissions [77]. In an effort to address these issues, Shanghai has embarked on initiatives to incorporate ISW into cement kilns. The Bailong Port area in Pudong, Shanghai, houses a large-scale solid waste comprehensive utilization demonstration base that utilizes cement kilns for the integrated treatment of urban solid

waste. With a planned annual disposal capacity exceeding 3 million tons, this facility processes 530,000 tons of sludge annually, with a moisture content of 80%. Preliminary data indicate that municipal sludge, household garbage, desulfurized gypsum, and fly ash, which are suitable for absorption, account for 53%, 15%, 33%, and 19% of Shanghai's current production, respectively. According to the on-site investigation, it was found that the use of the cement industry and cement kiln to co-dispose urban solid waste has unique advantages compared to traditional treatment methods. First of all, the cement industry can absorb a wide range of waste types, different types of waste can be used for cement industry as a mixture of materials, alternative raw materials or alternative fuels, so as to achieve the "waste into wealth". Secondly, the temperature in the cement kiln is much higher than that in the garbage incinerator. In such a state, the mass concentration of harmful components (such as PCDD/Fs and some emerging dioxin-like compounds) is significantly reduced when co-processing waste in cement kilns [78]. Finally, the heat energy generated by the combustion of combustible solid waste in the cement kiln can be directly used for the heat exchange process in the kiln system so as to reduce the energy consumption. Therefore, the use of cheap RDF in Shanghai's cement industry plays an important role in economic and environmental issues, helping to build a "zero-waste city" in Shanghai, and is worth promoting throughout the city.

## 4. Conclusions

This article presents a comprehensive review of research conducted in EU countries pertaining to the moisture content, ash content, calorific value, and elemental composition of RDF, as well as pretreatment technologies, additive selection, and flue gas control. Drawing upon this knowledge, the article puts forth a working framework and promising prospects for the development of RDF from LISW in China. The distinctive features of RDF hold significant relevance for developing countries like China as they contribute to sustainable development by reducing waste and mitigating environmental pollution. In comparison to China's national development policies, RDF emerges as an economically viable, practicable, and environmentally friendly alternative fuel. The large-scale utilization of RDF in China holds tremendous potential for environmental benefits and economic impact. To obtain high-quality RDF from ISW, several methods can be employed, including mechanical separation techniques (vibrating screens, air classification, magnetic separation) to remove impurities like organic waste, plastics, and metals, ultimately improving RDF quality. Additionally, biological conversion techniques can be used to convert organic waste into biomass fuel through biological degradation, further increasing the yield and quality of RDF. Thermal decomposition processes, involving high-temperature treatment of ISW, can transform it into combustible gases and residues, ultimately obtaining efficient RDF. In conclusion, by combining different technologies and methods, the goal of obtaining high-quality RDF from ISW can be achieved.

The future of RDF application on a global scale appears promising due to reliable modeling and simulation techniques, coupled with real-world industrial data. This is further driven by the urgency to address pressing environmental issues. However, challenges remain. Currently, RDF facilities are concentrated in developed nations. As developing countries experience economic and social growth, their municipal solid waste issues will also increase. To unlock RDF's full potential, further research is needed: investigating additional process parameters, exploring co-gasification with diverse feedstocks, and conducting optimization studies using modeling software. Additionally, life cycle assessments and circular economy considerations are crucial. Finally, sufficient funding is essential for widespread commercialization.

**Author Contributions:** Conceptualization, T.L. and W.L.; investigation, W.L.; resources, Z.L.; data curation, T.L.; writing—original draft preparation, T.L. and W.L.; writing—review and editing, T.L.; visualization, L.W. All authors have read and agreed to the published version of the manuscript.

**Funding:** This work was supported by Inner Mongolia Research Institute Project (SA1600213).

**Institutional Review Board Statement:** Not applicable.

**Informed Consent Statement:** Not applicable.

**Data Availability Statement:** No new data were created or analyzed in this study. Data sharing is not applicable to this article.

**Conflicts of Interest:** The authors declare no conflict of interest.

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
