# Peer review of "Comprehensive Analysis of Industrial Solid-Waste-to-Energy by Refuse-Derived Fuel Technology: A Case Study in Shanghai"

_sustainability, doi:10.3390/su16104234_

Round 1

Reviewer 1 Report

Comments and Suggestions for Authors

To: sustainability

From: Anonymous reviewer

Paper title: Comprehensive analysis of industrial solid waste-to-energy by RDF technology: A case study in Shanghai

Subject: sustainability-2870359-peer-review-v1

Date: 09.02.2024

A brief summary

The authors’ main goals were present work investigates possibilities of using industrial waste (ISW) after transforming it into refuse derived fuel (RDF) as fuel for power plants.  As an important alternative energy source and storage way, RDF was introduced and analyzed in details in this work. The advantages of RDF are mainly reflected in two aspects: management and technology. Management aspect takes into account: the regulatory aspects include the principles for the preparation of RDF raw materials, storage and transportation requirements, and the regulatory requirements for pollutant emissions. Technology aspects include pretreatment techniques, additive selection and analysis of RDF as an alternative to fossil fuels.Taking Shanghai as an example, we also analyzed the situation of ISW, domestic waste incineration plant and cement kiln plant, aim to find the application scenario of RDF in the future energy system. Taking into account that generation rate of ISW is rising and surpassing the natural environment capacity and the ability of municipal authorities to respond due to the development of technology and changes in consumption patterns, the topic of the manuscript is interesting for environmental reasons, obtaining alternative fuels.

However, submission contains technical and editorial drawbacks, which require some revision.

Broad comments

Introduction the introduction lacks a broader discussion of industrial solid waste, e.g. its composition, calorific value.

Status of RDF - without comment. Based on the literature data, authors have sufficiently discussed properties of RDF, principles of RDF storage, pollution control of RDF, technology of RDF manufacturing, pretreatment technology, choice of additives, application of RDF principle

Application of RDF principle - without comment.

The future of RDF in Shanghai - Please align the descriptions in Fig.4. In the figure description there is 4" and in the text 4R

Conclusion - in conclusion, the state of industrial development in Shanghai, which is linked to the increase in the amount of ISW.  What is lacking, however, is an indication of the methods to be followed to obtain RDF from the ISW.

References -  sufficient up-to-date literature was used in writing the paper.

Specific comments

The article provided for review is not formatted correctly. It consists of two parts: the first part contains an excerpt from the article from page 1 to page 7, while the full body of the article contains pages 8 to 15

Comments on the Quality of English Language

To: sustainability

From: Anonymous reviewer

Paper title: Comprehensive analysis of industrial solid waste-to-energy by RDF technology: A case study in Shanghai

Subject: sustainability-2870359-peer-review-v1

Date: 09.02.2024

A brief summary

The authors’ main goals were present work investigates possibilities of using industrial waste (ISW) after transforming it into refuse derived fuel (RDF) as fuel for power plants.  As an important alternative energy source and storage way, RDF was introduced and analyzed in details in this work. The advantages of RDF are mainly reflected in two aspects: management and technology. Management aspect takes into account: the regulatory aspects include the principles for the preparation of RDF raw materials, storage and transportation requirements, and the regulatory requirements for pollutant emissions. Technology aspects include pretreatment techniques, additive selection and analysis of RDF as an alternative to fossil fuels.Taking Shanghai as an example, we also analyzed the situation of ISW, domestic waste incineration plant and cement kiln plant, aim to find the application scenario of RDF in the future energy system. Taking into account that generation rate of ISW is rising and surpassing the natural environment capacity and the ability of municipal authorities to respond due to the development of technology and changes in consumption patterns, the topic of the manuscript is interesting for environmental reasons, obtaining alternative fuels.

However, submission contains technical and editorial drawbacks, which require some revision.

Broad comments

Introduction the introduction lacks a broader discussion of industrial solid waste, e.g. its composition, calorific value.

Status of RDF - without comment. Based on the literature data, authors have sufficiently discussed properties of RDF, principles of RDF storage, pollution control of RDF, technology of RDF manufacturing, pretreatment technology, choice of additives, application of RDF principle

Application of RDF principle - without comment.

The future of RDF in Shanghai - Please align the descriptions in Fig.4. In the figure description there is 4" and in the text 4R

Conclusion - in conclusion, the state of industrial development in Shanghai, which is linked to the increase in the amount of ISW.  What is lacking, however, is an indication of the methods to be followed to obtain RDF from the ISW.

References -  sufficient up-to-date literature was used in writing the paper.

Specific comments

The article provided for review is not formatted correctly. It consists of two parts: the first part contains an excerpt from the article from page 1 to page 7, while the full body of the article contains pages 8 to 15

Reviewer 2 Report

Comments and Suggestions for Authors

The manuscript titled "Comprehensive Analysis of Industrial Solid Waste-to-Energy by RDF Technology: A Case Study in Shanghai" addresses a critically important and timely topic in the field of waste management and sustainable energy. The challenge of managing industrial solid waste (ISW), particularly low-value industrial solid waste (LISW), is a significant environmental issue. The authors have undertaken an ambitious study to evaluate the potential of refuse-derived fuel (RDF) technology as a solution for the energy recovery and sustainable management of ISW in Shanghai, China's largest megalopolis.

The authors provide a thorough analysis of RDF's advantages, including management and technical aspects, and assess its potential as an alternative fuel source. By focusing on the specific context of Shanghai, the paper presents a compelling case for the viability and environmental benefits of RDF technology in managing LISW, highlighting its potential to contribute to a resource-saving and environment-friendly society in China.

Strengths:

  1. Relevance and Importance: The topic is highly relevant to current environmental and energy challenges, particularly in rapidly industrializing countries like China. The manuscript provides valuable insights into sustainable waste management practices.
  2. Comprehensive Analysis: The authors offer an in-depth examination of RDF technology, including its management and technical benefits, regulatory framework, and potential as an alternative energy source. This comprehensive approach strengthens the manuscript's contribution to the field.
  3. Case Study Approach: Using Shanghai as a case study adds specificity and relevance to the analysis, making the findings more tangible and applicable to similar urban and industrial contexts.

Recommendations for Revision:

While endorsing the manuscript for publication, I recommend minor revisions to enhance its clarity and compliance with journal standards:

  1. Eliminate Repetitive Paragraphs: Some sections of the manuscript appear to be repetitive. I suggest a thorough review to streamline the content, ensuring each paragraph contributes new information or perspectives to the discussion.
  2. Pagination and Formatting: The manuscript does not adhere to the journal's required template, particularly concerning pagination. Adjusting the format according to the journal's guidelines will aid in the manuscript's readability and professional presentation.
  3. Template Usage: The authors should revise the manuscript to fully comply with the journal's template, including headings, subheadings, and overall structure. This will facilitate a smoother review process and enhance the manuscript's alignment with journal standards.
  4. Figure Resolution: Figure 6 is of low resolution. Enhancing the figure's quality will ensure that all visual aids are clear and effectively contribute to the manuscript's arguments.

The manuscript provides a valuable contribution to the field of waste-to-energy technologies and sustainable waste management. Its focus on RDF technology as a solution for managing LISW in a densely populated urban environment is both pertinent and timely. With the suggested minor revisions, I believe the manuscript will make a significant impact on the field, offering useful insights and guidance for policymakers, researchers, and practitioners involved in waste management and sustainability initiatives. I endorse this article for publication following the recommended revisions. 

Reviewer 3 Report

Comments and Suggestions for Authors

The focus of the study is based on the use of RDF technology for treating industrial waste and energy generation.   Although the proposed idea is interesting, however, the manuscript lacks proper explanation and justification of the problem formulated and the method used.  The following limitations are observed in the manuscript;

1.        Literature review needs further enrichment in terms of developing research  gap in the domain of the undertaken study.

2.        The original contribution of the paper and its significance need to be clearly articulated. Why only RDF technology? Is there any other alternative to replace this idea? A critical explanation is required. 

3.        The problem to be addressed is not clearly formulated and needs further scholarly enhancement. Support your arguments from the scientific literature. 

4.        Discuss the clarity and appropriateness of the methodology used in the research.

5.        The conclusion of the paper briefly discussed which is the weakest part. This could involve discussing the practical implications of the research, highlighting any limitations or areas for future research, and reiterating the significance of the study's contributions.  Discussion is required on the significance of future directions in furthering the field of industrial waste treatment and energy generation through RDF technology. 

Comments on the Quality of English Language

Professional language editing is needed to improve the linguistic quality of the paper.

Round 2

Reviewer 3 Report

Comments and Suggestions for Authors

The authors have addressed the round 1 comments satisfactorily.